# An Online Preoperative Screening Tool to Optimize Care for Patients Undergoing Cancer Surgery: A Mixed-Method Study Protocol

**DOI:** 10.3390/cancers17050861

**Published:** 2025-03-03

**Authors:** Alexandria Paige Petridis, Cherry Koh, Michael Solomon, Sascha Karunaratne, Kate Alexander, Nicholas Hirst, Neil Pillinger, Linda Denehy, Bernhard Riedel, Chelsia Gillis, Sharon Carey, Kate McBride, Kate White, Haryana Dhillon, Patrick Campbell, Jack Reeves, Raaj Kishore Biswas, Daniel Steffens

**Affiliations:** 1Surgical Outcomes Research Centre (SOuRCe), Royal Prince Alfred Hospital, Sydney 2050, Australia; apet2033@uni.sydney.edu.au (A.P.P.); cherry.koh@sydney.edu.au (C.K.); professor.solomon@sydney.edu.au (M.S.); sascha.karunaratne@health.nsw.gov.au (S.K.); kate.alexander4@health.nsw.gov.au (K.A.); nicholas.hirst1@health.nsw.gov.au (N.H.); sharon.carey1@health.nsw.gov.au (S.C.); kate.mcbride@health.nsw.gov.au (K.M.); kate.white@sydney.edu.au (K.W.); pcam8352@uni.sydney.edu.au (P.C.); 2Faculty of Medicine and Health, Central Clinical School, The University of Sydney, Sydney 2050, Australia; neil.pillinger@sydney.edu.au; 3Institute of Academic Surgery (IAS), Royal Prince Alfred Hospital, Sydney 2050, Australia; 4Department of Anaesthesia, Perioperative Medicine, and Pain Medicine, Peter MacCallum Cancer Centre, Melbourne 3000, Australia; bernhard.riedel@petermac.org; 5Department of Health Services Research, Allied Health, Peter MacCallum Cancer Centre, Melbourne 3000, Australia; l.denehy@unimelb.edu.au; 6Department of Physiotherapy, Faculty of Medicine Dentistry and Health Sciences, The University of Melbourne, Melbourne 3010, Australia; 7The Sir Peter MacCallum Department of Oncology, and The Department of Critical Care, The University of Melbourne, Melbourne 3010, Australia; 8School of Human Nutrition, McGill University, Montreal, QC H9X 3V9, Canada; chelsia.gillis@mcgill.ca; 9Faculty of Science, School of Psychology, Centre for Medical Psychology & Evidence-Based Decision-Making, The University of Sydney, Sydney 2050, Australia; haryana.dhillon@sydney.edu.au; 10Graduate School of Health, Faculty of Health, University of Technology Sydney, Sydney 2007, Australia; jack.reeves@uts.edu.au; 11School of Health Sciences, Faculty of Medicine and Health, The University of Sydney, Sydney 2050, Australia; raaj.biswas@sydney.edu.au

**Keywords:** gastrointestinal cancer, surgery, screening tool, nutritional, physical, psychological

## Abstract

There is currently no standardised self-reporting tool, or established cut-off points for comprehensive risk assessment for patients undergoing cancer surgery. This study aims to develop, validate, and implement an online self-reporting preoperative screening tool that identifies modifiable risk factors. We believe that the timely identification of high-risk patients, based on their preoperative physical, nutritional, and psychological statuses observed through a self-reporting online screening tool, would streamline identification of high-risk patients, enabling referral to targeted interventions.

## 1. Introduction

Approximately 20% of the global population will develop cancer within their lifetime, with one in twelve people developing gastrointestinal cancer, and one in sixteen cases leading to death [1,2]. Colorectal cancer represents close to one in ten cancer cases and deaths, with the second highest rates of mortality. Liver cancer has the third highest rate of mortality and is the sixth most diagnosed cancer, while stomach cancer ranks fifth for both incidence and mortality worldwide [1]. According to the global cancer statistics for 2022, gastrointestinal cancer made up 24.6% of newly diagnosed cancers and 34.2% of cancer mortality [1]. The global demographic-based predictions indicate that the annual rate of cancer will increase by approximately 77% by 2050 [1].

Despite the rising incidence and advances in medical technology, surgery remains the mainstay of curative treatment, with or without chemotherapy and/or radiotherapy for selected patients [3]. The overall survival for patients undergoing gastrointestinal cancer surgery has improved significantly over the past two decades, with five-year survival rates ranging from less than 30% to 70% depending on disease stage [4]. However, rates of postoperative complications remain high, occurring in up to 65% of patients [5,6]. The presence of postoperative complications is associated with a greater burden to the patient, increasing hospital stay, decreasing quality of life, and increasing health service costs. Recent studies have demonstrated that patients presenting with poor preoperative physical status, nutrition, and/or mental health are up to 50% more likely to develop postoperative complications, subsequently having poor postoperative outcomes [7,8,9].

To attenuate the reduced physiological and functional capacity associated with gastrointestinal cancer surgery and the impact of preoperative comorbidities, recent studies have explored the role of prehabilitation in increasing physiological reserve and functional capacity to improve patient recovery and return to baseline functioning [10]. Gastrointestinal cancer carries a known physical, nutritional, psychological, and financial burden that may be increased with repeat admissions and longer hospital stay. A systematic review and meta-analysis performed by Mizrahi et. al. determined that on average, patients who underwent exercise prehabilitation reduced their length of hospital stay by approximately 1.4 days and reduced their hospital admission when compared to a usual care control [11]. Similarly, a systematic review and meta-analysis performed by Waterland et al. found a mean reduction in hospital length of stay of 3 days in abdominal cancer patients [12]. Studies focused on lung cancer populations have demonstrated a 4-8 day reduction in hospital length of stay [13,14]. Similarly, nutritional status has been shown to influence prehabilitation responses, with malnourishment being a barrier for improved functional capacity [15]. The identification of patients that are at a high risk of postoperative complications would allow for appropriate preoperative intervention selection. The establishment of a validated preoperative online screening tool, including measures of physical, nutritional, and psychological status, would enable tailored interventions, resulting in health optimisation before surgery and benefiting patients during the recovery period. Individual studies investigating physical, nutritional, and psychological status have shown the importance of positive preoperative states for improved recovery following colorectal surgery [16,17].

Currently, the main pitfall within the preoperative workup period is that the assessment of potential modifiable risk factors, such as physical status, nutrition, and mental health, is undertaken in the preadmission clinic, which normally takes place within one to two weeks of the planned surgical procedure [18]. While a team of doctors, nurses, and other health professionals conduct a comprehensive screening and assessment of patients at the preadmission clinics, most of the identified risk factors at this stage are either non-modifiable (e.g., diabetes mellitus) or are modifiable but are not able to be effectively addressed due to the limited time between the assessment and scheduled surgical procedure (e.g., deconditioning, malnutrition). Preadmission counselling involving patient education on surgical milestones and criteria for discharge have been suggested to enhance recovery following cancer surgery, resulting in decreased length of hospital stay and complication rates [19,20]. Early evaluation of patient limitations and expectations through a screening process can enable tailored interventions that patients may be more likely to adhere to. Thus, the innovative implementation of an online, self-reporting, preoperative screening tool 4–6 weeks prior to the surgery would be beneficial in the identification of modifiable risk factors that are associated with worse postoperative outcomes [21]. This can further be utilised to allow patients and clinicians to work collaboratively towards shared goals enabling enhanced recovery outcomes [22,23]. There are several available preoperative screening tools used to individually determine the status of patients before cancer surgery, including the following: (i) physical status: the Duke Activity Status Index (DASI) and the Community Health Activities Model Program for Seniors (CHAMPS), used to assess functional capacity [10,24]; (ii) nutrition: the Patient-Generated Subjective Global Assessment-Short Form (PG-SGA-SF), Nutrition Risk Screening (NRS), and Malnutritional Universal Screening Tool (MUST) have proven their ability to identify adverse and enhance clinical outcomes in cancer patients [20,25,26]; and (iii) psychological tools: the Hospital Anxiety and Depression Scales (HADS), Kessler Psychological Distress Scale (K6), and Patient Health Questionnaire depression scale (PHQ-9) [7,27,28].

Despite the availability of these tools and the strong correlations demonstrated with gold standard objective measures, none of the identified tools take into consideration all three domains (i.e., physical, nutritional, and psychological). Currently, there is no standardised screening tool or recommended optimal cut-off point that can be used to identify individuals at higher risk of poor outcomes when undergoing gastrointestinal cancer surgery, as most studies use different tools. Therefore, the main aim of this study is to develop, test, and implement a simple, online, self-reporting, preoperative risk stratification tool that will identify modifiable physical, nutritional, and/or psychological factors to ensure patients receive targeted and tailored interventions. Hence, the specific aims of this project are to (i) develop a preoperative online tool to assess self-reported measures of physical capacity, nutrition, and mental health, using previously validated questionnaires in accordance with experts’ feedback; (ii) investigate the potential association between the preoperative online tool and postoperative surgical outcomes, including postoperative complications and length of stay, and to determine risk stratification thresholds for the preoperative online tool; and (iii) to implement the preoperative online tool in the preoperative hospital setting.

## 2. Materials and Methods

### 2.1. Study Design

This project utilises a rigorous sequential exploratory mixed-methods design, incorporated within three distinct stages (Figure 1) [29].

### 2.2. Stage 1—Development

The preoperative online screening tool will be developed according to three research streams, as below.

#### 2.2.1. Identification of Preoperative Screening Tools

##### Protocol and Registration

A scoping review will be performed according to the methodological framework proposed by Arksey and O’Malley [30], and reported according to the Preferred Reporting Items for Systematic Reviews and Meta-Analysis (PRISMA) [31,32], with the detailed review protocol published a priori on the Open Science Framework platform. The main aim of this scoping review is to identify the most up-to-date tools suitable to be used in the preoperative period to screen physical, nutritional, and/or psychological factors in surgical cancer patients.

##### Search Strategy and Eligibility

A search strategy will be created in conjunction with an experienced librarian from The University of Sydney and performed on the Medline, Embase, Evidence Based Medicine (EBM), Cumulative Index of Nursing and Allied Health Literature (CINAHL), and PsycINFO databases. A combination of the following keywords and subject headings will be used: ‘screening tools’ AND ‘gastrointestinal neoplasms’ AND ‘preoperative’ AND ‘nutrition’ OR ‘psychological’ OR ‘exercise’. The completed search strategy will be made available upon completion. The bibliographies of included studies found through the database search will also be examined. Studies published in the literature after the year 2000, written in the English language, including adult (≥18 years) patients undergoing gastrointestinal cancer surgeries and reporting on physical, nutritional, and/or psychological screening tools will be included. Gastrointestinal surgeries on the liver, stomach, spleen, pancreas, peritoneal, colon, rectum, and oesophagus will be included.

Systematic reviews and meta-analyses, randomised control trials (RCTs), retrospective and prospective cohort studies, cross-sectional studies, and case series with >10 participants undergoing gastrointestinal cancer surgery will be included. Conference abstracts will also be examined. Studies reporting screening tools on mixed populations (>80% of the sample non-cancer) or patients with any cancer not listed above will be excluded.

##### Selection and Data Collection Process

Two review authors will independently screen all the articles and extract all the data. As this is a scoping review, we are not planning to assess the quality of the identified articles. The COVIDENCE software will be used to manage the review and synthesise the results [33]. Detailed information on the identified screening tools will include screening tool name, year developed, recall period, number of items, number of domains, response options, score range, time to complete, tool availability, use in cancer surgical patients, and psychometric information. This review will provide invaluable information for the first round of Section 2.2.2. (below).

#### 2.2.2. Delphi Study

##### Study Design

The Delphi methodology will be employed for gaining consensus on the selection of the most appropriate preoperative physical, nutritional, and psychological screening tools for patients undergoing elective gastrointestinal cancer surgery. Each identified tool will be grouped by domain (physical, nutritional, and psychological) before undergoing a three-round iterative survey.

##### Participants, Recruitment, and Setting

Guidelines proposed by Hasson and Colleagues [34] and the recommendations from Conducting and REporting of DElphi Studies (CREDES) [35] will be used to establish consensus across a multidisciplinary panel of local, national, and international experts including doctors, nurses, allied health professionals, research groups, statisticians, epidemiologists, and health- and policymakers. This approach will facilitate the representation of professional groups that directly influence patient decisions and would benefit from the results of this project. For this, members of the Global (P)rehabilitation Initiative (https://source.sydney.edu.au/global-prehabilitation-initiative/ (accessed on 12 May 2024)), consisting of approximately 2000 multidisciplinary experts from 6 continents, will be approached and invited to participate. In addition, authors from the identified literature (Section 2.2.1) will also be invited to participate. The invitation letter will also ask individuals to disseminate the study invitation to their networks. Consent will be gained by all participants wanting to form part of the Delphi survey, and ethical consultation will be sought from Sydney Local Health District Human Research Ethics Committee (HREC). It is estimated that between 100 and 200 participants will form the multidisciplinary panel involved in this Delphi survey.

Demographic characteristics including age, gender, education level, role/discipline, country of practised, and involvement within prehabilitation will be collected from the expert panel. A three-step Delphi method will be employed through a secure web application (REDCap) [36].

### 2.3. Delphi Methodology

#### 2.3.1. First Round

The preoperative screening tools identified in the abovementioned scoping review (Section 2.2.1) will be compiled in the first round and distributed via email to participants. In this round, experts will have the opportunity to add extra preoperative screening tools there were not identified in the scoping review. Identified tools will be categorised based on the factors they encompass (i.e., physical, nutritional, psychological). Using a 5-point Likert Scale (1 = no importance, 2 = limited importance, 3 = neither important nor unimportant, 4 = somewhat important, 5 = critically important), determination of the appropriateness of tools can be made. Furthermore, we will be able to rate identified screening tools based on the following consensus criteria prior to Round 2 [37,38]:Consensus in: ≥ 70%of experts on the use of appropriate tools, marking them as important (≥4).Consensus out: ≥ 70%of experts on the dismissal of tools, marking them as of limited importance (≤2).No consensus: neither of the above criteria are met.

#### 2.3.2. Second Round

In Round 2, experts will be emailed an updated list of preoperative screening tools with the group scores and their individual scores. The abovementioned Likert scale will be implemented to allow for newly identified tools to be compared to the same standards previously used. Bias will be mitigated by randomisation of tools, and properties identified by 70% of participants as either important or critically important, or without consensus, will be collated for the third round.

#### 2.3.3. Third Round

Round 3 will be comprised of a final Delphi survey; using the abovementioned Likert scale, members will be encouraged to discuss the remaining preoperative screening tools until agreement is reached to retain, modify, or eliminate from the final statement. Following Round 3, a consensus on appropriate tools of ≥70% and a median score of ≥4 on the Likert scale will be used as criteria to form the online screening tool (Section 2.4). If neither criteria 1 or 2 is met at the completion of three rounds, consensus will be defined as the majority agreement [39].

#### 2.3.4. Statistical Analysis

Descriptive statistics will be employed to summarise participant characteristics. All three rounds will calculate the median scores of each identified preoperative screening tool using the 5-item Likert Scale previously described, and the inter-rater reliability (ICC) will be calculated.

### 2.4. Development of Online Screening Tool

This step will utilise the evidence acquired in Section 2.2.2 to develop a secure web application that will allow the management of the online preoperative screening tool and data collection. Original developers of the identified screening tools will be consulted regarding their use of the online screening tool. For tools that are not developed in an online survey format (i.e., solely paper-based tools or those using consultation/telephone), or not freely available, developers will be approached regarding use within this online screening tool. We will assemble a focus group including clinicians and experts, along with patients and health consumers, to determine the format of the online preoperative screening tool. This will include pilot testing to check the comprehensibility and useability of the tool. The finalised tool will be used in stage 2 below.

### 2.5. Stage 2—Testing

#### 2.5.1. Design and Setting

A prospective longitudinal cohort study will be conducted by recruiting consecutive patients undergoing elective gastrointestinal cancer surgery at several national and international hospitals over a 12-month period beginning at the completion of the Delphi survey. All patients will be required to complete the informed consent form and will be followed until 30 days post-operation. Patient data will be collected from the online screening tool and electronic medical records over this period. This study protocol was written in accordance with the STROBE statement [40].

#### 2.5.2. Recruitment

A range of national and international centres will be approached, with surgeons, anaesthetists, and clinical researchers from the participating hospitals involved in patient screening. All patients that meet our inclusion criteria will be provided with an information sheet regarding the study, and if interested in participating, a researcher from the study will contact them to discuss the study further. Clinical assessment and decision-making will proceed according to usual clinical protocols.

#### 2.5.3. Participants

All participants must satisfy the following inclusion criteria and none of the exclusion criteria:

#### 2.5.4. Inclusion Criteria

Adults (aged ≥ 18 years) of any gender;Patients who are undergoing elective curative surgery for gastrointestinal cancer including for their colon, rectum, pancreas, stomach, liver, spleen, or oesophagus;Patients with the capacity to provide informed consent;

#### 2.5.5. Exclusion Criteria

Patients who have inadequate English language skills to complete the online screening tool or are unable to be assisted with the completion of the online screening tool.

#### 2.5.6. Participant Consent

All participants will be asked to complete a written consent form after they have read the provided information sheet and have had time to consider participation. Interested participants will be provided with a secure link to the preoperative screening tool. Participants involved are under no obligation to complete the screening tools and can withdraw from the study at any time. Participants can be assured that choosing to withdraw from the study will not affect any relationships with their treating team or quality of care that will be received.

#### 2.5.7. Outcome and Measurements

Consenting patients will be emailed a secure link containing the online preoperative screening tool (Section 2.4) during their visit to the surgical clinic, usually 4–6 weeks prior to their scheduled procedure. Non-responders will be sent reminder emails within 7–14 days to maximise the response rate. Participants will be included if they complete the screening tool between 1 and 12 weeks prior to their scheduled surgery.

All screening tool data will be collated, including the patient characteristics and time between survey and surgical procedure, to gather concise information regarding patient characteristics and feasibility.

#### 2.5.8. Sample Size Calculation

A minimum sample size of 200 patients is required to determine the correlation between the preoperative screening tool scores and postoperative complications (primary outcome), assuming a moderate effect size (r = 0.3), α = 0.05, and 80% power, while accounting for potential dropouts (30%) and multiple variables in the logistic regression analysis. However, we aim to recruit a larger sample to enable robust determination of risk thresholds for individual patient cohorts based on cancer type (e.g., colorectal, pancreatic, gastric), patient characteristics (e.g., age groups, comorbidity profiles), and surgical approaches.

#### 2.5.9. Surgical Outcomes

All outcomes will be independently extracted from the hospital electronic medical records by study personnel blinded to the patients’ preoperative status.

#### 2.5.10. Primary Outcomes

Proportion of patients developing in-hospital postoperative complications: defined as any deviation from the normal postoperative course and classified according to the Clavien–Dindo Classification (CDC) [41].

#### 2.5.11. Secondary Outcomes

Intensive Care Unit (ICU) length of stay (days);Hospital length of stay (days);All-cause perioperative mortality (up to 30 days from index surgery).

Other patient characteristics and surgical outcomes extracted will include age, gender, type of cancer, surgical procedure, surgical time, discharge destination, and hospital readmission within 30 days. Acceptability of the preoperative online screening tool and willingness to participate in non-routine preoperative interventions will also be measured.

Other outcomes will include the Comprehensive Complications Index (CCI) [42,43] and Days Alive and at Home within 30 days (DAH30) [44,45]. The Comprehensive Complications Index (CCI) uses complications graded with the CDC, where each CDC grade has a specific CCI score and weight of complication (wC), to provide an overall accumulating score between 0 (no complications) and 100 (death) [42]. CCI score can be generated using the freely available online tool at https://www.cci-calculator.com/ (accessed on 12 May 2024) or calculated using the following formula:
$$CCI=\frac{\sqrt{wCI+wC2\ldots +wCx}}{2}$$


DAH30 will be calculated using mortality and hospitalisation data from the date of the index surgery, as described in detail by Myles et al. [44].

#### 2.5.12. Data Collection and Analyses

All data will be collected through the widely available, secure REDCap data management platform [36]. Descriptive statistics will be used to describe the sample characteristics and outcome measures. The association between the preoperative screening tool scores and postoperative outcomes will be assessed in a logistic regression analysis. The risk stratification thresholds of the preoperative online screening tool will be derived from a receiver operating characteristic analysis. All analysis will be conducted in SPSS version 29, with *p* < 0.05 indicating statistical significance. A comprehensive statistical analysis plan will be developed by an experienced statistician prior to the start of the study.

### 2.6. Stage 3—Implementation and Sustainability

The implementation of the preoperative online screening tool is embedded within the project pathway. Our careful selection of project investigators is one key strategy to achieve this. Healthcare professionals involved in the multidisciplinary panel will develop communication strategies that can aid with implementation in clinical practise and sustainability plans. Their strengths include renowned multidisciplinary clinical expertise, active presence and leadership in cancer team meetings, influence on treatment decision-making, experience in bedside-to-bench translation research, and strong involvement in investigator-initiated implementation strategies. They are well positioned to influence its uptake in their immediate clinical environments. Together, they will prepare an implementation policy document. The implementation policy will consider the feasibility of the web-based application, including, but not limited to, time taken to complete online screening, the need for additional support to patients, follow-up, how it can be implemented into practise with minimal implications for workflow, necessary staff, and patient satisfaction [46]. A staggered and reiterative process for the implementation of web-based health interventions, such as this, allows the intervention to be tailored to address the underlying multifaceted social, technological, and organisation factors [47].

Potential barriers to implementation must be considered to ensure successful adoption and sustainability. Resistance to change among healthcare professionals and patients may pose a significant challenge. Technical issues, such as software bugs or user interface problems, could hinder the tool’s effectiveness. Resource constraints, including limited staff and financial resources, may also impede the implementation process. Ensuring data privacy and compliance with relevant regulations is essential to protect patient information. Integrating the tool seamlessly into existing clinical workflows requires careful planning and training. Additionally, maintaining the tool’s relevance and effectiveness over time is essential for sustainability

Building on implementation science principles, we will utilise the Triple C framework (Consultation, Collaboration, and Consolidation) [48]. The consultation phase will engage stakeholders in identifying priorities and mapping processes. Healthcare professionals will participate in multidisciplinary planning sessions to establish implementation pathways. The collaboration phase will coordinate across departments and specialties, establishing clear communication channels and knowledge sharing protocols. The consolidation phase will focus on standardising procedures, monitoring outcomes, and ensuring sustainability through defined metrics.

To ensure it is widely available and readily accessible, the preoperative online screening tool will be made available on our website and published via peer-review article. We will work on communications strategies to ensure that health professionals are aware of the tool, and its benefits and ease of use. Informing patients on the use of this questionnaire at the time of, or prior to, appointments will allow for any queries or concerns to be addressed and managed.

We will seek support from the participating hospitals’ chief-executive for the sustainability of the preoperative online screening tool by implementing the tool into clinical practise and using the data generated to update the tool. Finally, the tool will be added to relevant teaching units within several universities’ health degree programmes to encourage its use by future medical and other health graduates, and consumer organisations.

## 3. Conclusions

Cancer is one of the leading causes of premature death worldwide. While surgery is the mainstay treatment option for gastrointestinal cancers, high rates of postoperative complications are associated with poorer prognosis [49]. Identifying patients at high risk of adverse postoperative outcomes before elective cancer surgery is challenging due to the short time between the preoperative work-up period and surgery. Therefore, a preoperative, self-reporting online screening tool that could be used by patients at home, early in the preoperative period (4–6 weeks prior), that will allow stratification of patient risk and identify the need for non-routine preoperative physical, nutritional, and/or psychological intervention would be extremely beneficial. This, in turn, has the potential to improve patient health outcomes, lower treatment costs, and provide equity to preoperative treatment optimisation. Prior to the conceptualisation of this project, our team conducted a rapid review of the literature. While we have identified some studies investigating preoperative risk assessment tools [50], most were for non-cancer or older patients [3], and were not online or focused on modifiable physical, nutritional, or psychological factors. We were surprised to find that there is currently no available preoperative online screening tool to identify patients (according to their preoperative physical, nutritional, and psychological statuses) at high risk of having worse surgical outcomes (postoperative complications and longer length of hospital stay). Therefore, we believe that this project will close this evidence gap by developing, testing, and implementing an online preoperative screening tool for patients undergoing cancer surgery. The identification of high-risk patients, based on their preoperative physical, nutritional, and psychological statuses, could benefit patients by allowing referral to non-routine interventions. Currently, the referral for these non-routine interventions is inconsistent, primarily based on individuals and each centres’ experiences. Often, they do not follow a comprehensive evidence-based approach that is supported by experts and consumers. The use of a preoperative online screening tool could minimise unwarranted variation in preoperative treatment optimisation. Furthermore, preoperative risk stratification by a self-reporting online screening tool could help allow for the safe provision of surgery to low-risk patients.

## 4. Ethics and Dissemination

Stage 1 of the protocol involves a scoping review of the available literature, a Delphi survey study involving a multidisciplinary panel, and the formulation of a secure web-based online screening tool. Ethics approval will be sought from Sydney Local Health District HREC and site-specific authorisation for participating centres in the Delphi study.

Stage 2 consists of a prospective cohort study in which ethics approval will again be sought from the Sydney Local Health District HREC, with site-specific authorisation sought from participating centres, and a data transfer agreement stating that each site will provide data to the REDCap system [36]. Stage 3 will be further discussed in an implementation protocol upon completion of first two stages, with ethical consultation sought at this time.

Results from all stages of this study will be submitted for publication in scientific journals and presented at research conferences and scientific meetings.

## Figures and Tables

**Figure 1 cancers-17-00861-f001:**
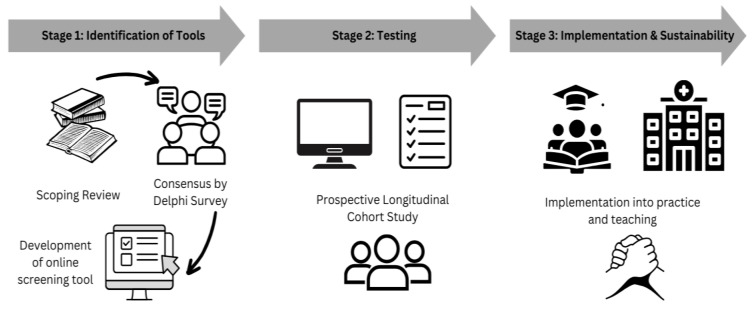
Schematical representation for development of online preoperative screening tool for cancer patients undergoing gastrointestinal surgery.

## Data Availability

No new data were created or analysed in this study. Data sharing is not applicable to this article.

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
