# Peer review of "An Online Preoperative Screening Tool to Optimize Care for Patients Undergoing Cancer Surgery: A Mixed-Method Study Protocol"

_cancers, 2025, doi:10.3390/cancers17050861_

Round 1

Reviewer 1 Report

Comments and Suggestions for Authors

I have read and reviewed the manuscript “Online preoperative screening tool to optimize care for patients 4 undergoing cancer surgery: A mixed method study protocol”. This is an interesting proposal that I look forward to eventually seeing the results in a few years. There are some issues with citations, I hope that my points below are clear. Also, a bit more info about inclusion/exclusion criteria would be beneficial to narrow the testing cohort (see points below). Finally, the “Minor Points” below are just suggestions for the authors to clarify a bit of the writing. Good luck on this project!

Major Points:

  1. The statement at line 93 that relates to lung cancer cites the paper by Mrzhari et al (reference 11). However, that exact point is actually not an independent conclusion from that paper and is actually comes from two separate papers (Sekine et al (Jpn J Thorac Cardiovasc Surgery 2005) and Pehlivan et al (Ann Thorac Cardiovasc Surg 2011). It is important to cite primary sources and not cite a paper that cites someone else. On that note, please ensure that the statement “approximately two days and reduced their hospital readmission by approximately 10%, when compared to a usual care control” is also from the original primary source, unless this is the finding of the meta-analysis.
  2. The same point about the sentence “Other prehabilitation reviews focused on lung cancer populations have demonstrated a 4-day reduction on length of hospital stay”. Since reference 11 is cited 3 times in 3 sentences, it is not “other prehabilitation reviews”. Please provide the appropriate citations for the “other prehabilitation reviews”.
  3. I agree completely with the authors that an evaluation “4-6 weeks prior to surgery” would be beneficial. However, sometimes this is not possible. Do the authors recommend getting their assessment and delaying patients at “high risk”? How might this impact operability or outcomes? How do the authors suggest getting these “pre-evaluations” done in a timely manner that allows for intervention?
  4. What is the inclusion criterion (section 1.3, stage 2) for the time range prior to surgery to complete the survey? It is stated that “usually 4-6 weeks prior (line 304). But, this should specifically defined with a window. Would they exclude an esophageal cancer patient that is seen prior to induction therapy (which is about 3 months prior to surgery)? Or, would they wait until after induction therapy to do the survey? These are just scenarios to consider. Also, is there an exclusion? (cannot be done within a week of surgery) Also, if this is only for elective surgeries, this should be in the inclusion criteria (or state that urgent/emergent surgeries will be excluded.) This should probably ONLY include elective operations.
  5. A power analysis to estimate the number of patents required to demonstrate validity of this tool is needed. This is important as not every patient who is approached will enroll.

Minor Points:

  1. In the first line of the “Simple Summary” might it be of benefit to state that there is “no standardized self-reported tool, or established cut-off points for comprehensive risk assessment” for patients undergoing cancer surgery? This is the thrust of the paper. Then, of course, a slight edit is needed in the second sentence to not duplicate the statement.
  2. Line 72, what is “mean surgery”? Consider an edit: Despite the rising incidence and advances in medical technology, surgery remains the mainstay of curative treatment, with or without chemotherapy and/or radiation therapy, for selected patients.
  3. Does the survival not only depend on of the primary tumor type in addition to stage? Consider revision.

Author Response

Dear Reviewer, thank you for taking the time to review my article.
below I have responded to your comments, please let me know if you feel something hasn't been addressed adequately or if I can further improve.

Comments 1: The statement at line 93 that relates to lung cancer cites the paper by Mrzhari et al (reference 11). However, that exact point is actually not an independent conclusion from that paper and is actually comes from two separate papers (Sekine et al (Jpn J Thorac Cardiovasc Surgery 2005) and Pehlivan et al (Ann Thorac Cardiovasc Surg 2011). It is important to cite primary sources and not cite a paper that cites someone else. On that note, please ensure that the statement “approximately two days and reduced their hospital readmission by approximately 10%, when compared to a usual care control” is also from the original primary source, unless this is the finding of the meta-analysis.

Response 1: Thank you for pointing this out, I agree with both of these statements. I have edited line 97 to better reflect the findings of the Meta-analysis “1.4 days and reduced hospital admission by 8%”. I have also changed the statement regarding lung cancer populations to include the original authors.

Comments 2: The same point about the sentence “Other prehabilitation reviews focused on lung cancer populations have demonstrated a 4-day reduction on length of hospital stay”. Since reference 11 is cited 3 times in 3 sentences, it is not “other prehabilitation reviews”. Please provide the appropriate citations for the “other prehabilitation reviews”.

Response 2: Thank you for highlighting this, I agree this was an error that should have been adjusted. Similarly to the above point, I have amended this paragraph to better encompass the results as a whole (Line 95-101). “A systematic review and meta-analysis performed by Mizrahi et. al., determined that on average, patients who underwent exercise prehabilitation reduced their length of hospital stay by approximately 1.4 days and reduced their hospital admission when compared to a usual care control. Similarly, a systematic review and meta-analysis performed by Waterland et. al, determined that, a mean reduction in hospital length of stay of 3 days in abdominal cancer patients. Studies focused on lung cancer populations have demonstrated a 4-8 day reduction in hospital length of stay”

.

Comment 3: I agree completely with the authors that an evaluation “4-6 weeks prior to surgery” would be beneficial. However, sometimes this is not possible. Do the authors recommend getting their assessment and delaying patients at “high risk”? How might this impact operability or outcomes? How do the authors suggest getting these “pre-evaluations” done in a timely manner that allows for intervention?

Response 3: You raise a valid concern about the feasibility of the 4-6 week preoperative assessment timeframe. The protocol currently positions this timing within the context of identifying modifiable risk factors that could benefit from intervention. As noted in the Introduction, the main pitfall in current practice is that assessment typically occurs in preadmission clinics 1-2 weeks before surgery, when many risk factors are either non-modifiable or cannot be effectively addressed due to time constraints.

The protocol's implementation strategy (Stage 3) acknowledges the need for flexibility in clinical workflows. Through the Triple C framework (Consultation, Collaboration and Consolidation), we will work with stakeholders to establish implementation pathways that account for real-world timing constraints while maximizing opportunities for intervention. This includes developing clear communication strategies and standardized procedures that can be integrated into existing surgical pathways.

Comment 4: What is the inclusion criterion (section 1.3, stage 2) for the time range prior to surgery to complete the survey? It is stated that “usually 4-6 weeks prior (line 304). But, this should specifically defined with a window. Would they exclude an esophageal cancer patient that is seen prior to induction therapy (which is about 3 months prior to surgery)? Or, would they wait until after induction therapy to do the survey? These are just scenarios to consider. Also, is there an exclusion? (cannot be done within a week of surgery) Also, if this is only for elective surgeries, this should be in the inclusion criteria (or state that urgent/emergent surgeries will be excluded.) This should probably ONLY include elective operations.

Response 4: Thank you for highlighting the need to clarify the timing criteria for the preoperative screening tool. You raise important points about the inclusion window and various clinical scenarios. We will revise the protocol to specify that patients will be included if they complete the screening tool between 1 to 12 weeks prior to their scheduled surgery (Line 331). This broad window is intentionally selected as the primary aim of this study is to identify risk thresholds within the selected screening tools that correlate with postoperative outcomes. As stated in our specific aims, we are focused on: (i) developing the preoperative online tool using validated questionnaires, (ii) investigating associations between the tool's results and postoperative outcomes, and (iii) implementing the screening tool in the preoperative setting.

The determination of optimal timing and the number of intervention sessions needed to enhance patient health before surgery, while important, falls outside the scope of this current project. These aspects would be valuable considerations for future research once the risk thresholds have been established through this study.

Comment 5: A power analysis to estimate the number of patents required to demonstrate validity of this tool is needed. This is important as not every patient who is approached will enroll.

Response 5: Thank you for highlighting the need for a clear power analysis. You raise an important point about sample size requirements, particularly for subgroup analyses. We will add a "Sample Size Calculation" subsection under "Stage 2 - Testing" in the Methods section: ‘A minimum sample size of 200 patients is required to determine the correlation between the preoperative screening tool scores and postoperative complications (primary outcome), assuming a moderate effect size (r=0.3), α=0.05, and 80% power, while accounting for potential dropouts (30%) and multiple variables in the logistic regression analysis. However, we aim to recruit a larger sample to enable robust determination of risk thresholds for individual patient cohorts based on cancer type (e.g., colorectal, pancreatic, gastric), patient characteristics (e.g., age groups, comorbidity profiles), and surgical approaches.’ (Line 337)

This expanded recruitment will strengthen the tool's validity across diverse patient populations and clinical scenarios. The sample size will be monitored during recruitment, with the possibility of extending the recruitment period to achieve adequate numbers for these subgroup analyses.

Minor Points:

Comment 6: In the first line of the “Simple Summary” might it be of benefit to state that there is “no standardized self-reported tool, or established cut-off points for comprehensive risk assessment” for patients undergoing cancer surgery? This is the thrust of the paper. Then, of course, a slight edit is needed in the second sentence to not duplicate the statement.

Response 8: Thank you for this suggestion, I agree this could have been changed to be more succinct. This has been edited appropriately in line 30.

Comment 7: Line 72, what is “mean surgery”? Consider an edit: Despite the rising incidence and advances in medical technology, surgery remains the mainstay of curative treatment, with or without chemotherapy and/or radiation therapy, for selected patients.

Response 7: Thank you for pointing this out, I have made the appropriate edits to make this statement clearer.

Comment 8: Does the survival not only depend on of the primary tumor type in addition to stage? Consider revision.

Response 8: I agree with this statement that survival is dependent on both tumor type and stage. I apologize, I am unsure of what you revision you are suggesting with this comment.

Hope you have a lovely day

Reviewer 2 Report

Comments and Suggestions for Authors

Dear author,

Many thanks for submitting your work to the journal. Some issues should be addressed.

Your study is well-structured, utilizing a mixed-methods approach that includes a scoping review, Delphi consensus, and a prospective cohort study to develop and validate an online preoperative screening tool. However, some issues have emerged and should be addressed:

  • The introduction provides data on cancer surgery risks and prehabilitation but lacks a focus on how existing tools fail to provide comprehensive screening. I suggest reorganizing this section to first describe the problem, then the current screening limitations, and finally how this study intends to address the gap.

  • Provide more details on the selection criteria for participants in the Delphi panel. Also, include considerations for potential biases in expert selection.

  • The study does not include a control group receiving standard assessment without the tool. Consider a control group in the testing phase.

  • The study aims to develop risk stratification thresholds using ROC analysis; however, the method of setting cut-off points is not clear. Please define cut-offs.

  • Will patients with low digital literacy or limited internet access be excluded from the screening?

  • There is limited discussion of potential issues in implementation. Please address potential barriers and strategies for preventing them.

  • Conclusions overstate the effectiveness of the tool without presenting the need for further validation. Further validation in diverse settings will be required.

Best regards

Author Response

Dear reviewer, thank you for taking the time to revise our protocol.

Comment 1: The introduction provides data on cancer surgery risks and prehabilitation but lacks a focus on how existing tools fail to provide comprehensive screening. I suggest reorganizing this section to first describe the problem, then the current screening limitations, and finally how this study intends to address the gap.

Response 1: Thank you for your review of this protocol. The introduction begins with an overview of the high incidence and mortality rates associated with gastrointestinal cancers and the significant post-operative complications. It then addresses the limitations of current screening tools in comprehensively assessing physical, nutritional, and psychological factors simultaneously. Finally, it outlines the intention to develop a new preoperative screening tool to fill this gap. Could you please provide further guidance on where additional evaluation is needed so that I may make the necessary adjustments?

Comment 2: Provide more details on the selection criteria for participants in the Delphi panel. Also, include considerations for potential biases in expert selection.

Response 2: We are inviting all members of the global prehabilitation network as well as all authors from the identified literature to participate in the Delphi panel. This approach includes those who both support and challenge the validity of each tool, ensuring a diverse range of responses. The only exclusion criterion is only non-participation in the survey.

Comment 3: The study does not include a control group receiving standard assessment without the tool. Consider a control group in the testing phase.

Response 3: Thank you for bringing up this point. The testing phase uses a longitudinal cohort study design, including all participants who meet the inclusion criteria, rather than comparing separate groups. The objective is to evaluate whether our screening tool can differentiate within the cohort that has completed the survey. Adding a control group without the survey may provide fewer details about whether our screening tool can accurately identify those at risk. Could you please provide more details on your perspective regarding the use of a control group in this testing phase so we can better consider its potential implementation?

Comment 4: The study aims to develop risk stratification thresholds using ROC analysis; however, the method of setting cut-off points is not clear. Please define cut-offs.

Response 4: Thank you for your review, it is correct that risk stratification thresholds will be developed using ROC analysis. The cut-off points will be determined based on the optimal balance of sensitivity and specificity and will be clearly defined in the study protocol. The development of this analysis plan will occur at a later date, once the study is initiated using an experienced statistician. I apologize that at this stage, we are unable to define further as this is something that requires the completion of stage 1 prior to initiation of stage 2.

Comment 5: Will patients with low digital literacy or limited internet access be excluded from the screening?

Response 5: Participants with low digital literacy or limited internet access will not be excluded from this study as we will be providing them with a secure link during their visit to the surgical clinic (line 331). By doing so, this will enable those who are interested but are unable to independently complete the survey to be assisted during this time (line 442). This will be further expanded within the future cohort study protocol.

Comment 6: There is limited discussion of potential issues in implementation. Please address potential barriers and strategies for preventing them.

Response 6: Thank you for pointing this out. We have now included a detailed discussion on potential implementation issues of the preoperative online screening tool within the protocol (line 409-417). This includes identifying barriers such as digital literacy, access to technology, and compliance, as well as strategies to mitigate these challenges. By addressing these potential issues, we aim to ensure that the tool is accessible and effective for all participants.

Comment 7: Conclusions overstate the effectiveness of the tool without presenting the need for further validation. Further validation in diverse settings will be required.

Response 7: Thank you for your insightful feedback. We acknowledge that while our initial findings are promising, further validation in diverse settings is crucial to fully establish the tool's effectiveness. Our rapid review highlighted a significant gap in the availability of preoperative online screening tools for cancer patients, particularly those focusing on modifiable physical, nutritional, and psychological factors. This project aims to address this gap by developing, testing, and implementing such a tool.

We agree that additional testing across various settings and populations is necessary to ensure the tool's reliability and generalizability. We have adjusted the conclusion to try to mitigate the overstatement. Please let me know if there is more you feel can be adjusted.

Please let me know if there are any further changes I may make to further improve.

Best wishes